# Novel Antibody–Peptide Binding Assay Indicates Presence of Immunoglobulins against EGFR Phospho-Site S1166 in High-Grade Glioma

**DOI:** 10.3390/ijms23095061

**Published:** 2022-05-02

**Authors:** Lona Zeneyedpour, Christoph Stingl, Johan M. Kros, Peter A. E. Sillevis Smitt, Theo M. Luider

**Affiliations:** 1Department of Neurology, Erasmus MC, 3015 GD Rotterdam, The Netherlands; l.zeneyedpour@erasmusmc.nl (L.Z.); c.stingl@erasmusmc.nl (C.S.); p.sillevissmitt@erasmusmc.nl (P.A.E.S.S.); 2Department of Pathology, Erasmus MC, 3015 GD Rotterdam, The Netherlands; j.m.kros@erasmusmc.nl

**Keywords:** antibodies, tumor-specific antigen, Melon Gel, phospho-proteomics, mass spectrometry

## Abstract

We investigated the feasibility of detecting the presence of specific autoantibodies against potential tumor-associated peptide antigens by enriching these antibody–peptide complexes using Melon Gel resin and mass spectrometry. Our goal was to find tumor-associated phospho-sites that trigger immunoreactions and raise autoantibodies that are detectable in plasma of glioma patients. Such immunoglobulins can potentially be used as targets in immunotherapy. To that aim, we describe a method to detect the presence of antibodies in biological samples that are specific to selected clinically relevant peptides. The method is based on the formation of antibody–peptide complexes by mixing patient plasma with a glioblastoma multiforme (GBM) derived peptide library, enrichment of antibodies and antibody–peptide complexes, the separation of peptides after they are released from immunoglobulins by molecular weight filtration and finally mass spectrometric quantification of these peptides. As proof of concept, we successfully applied the method to dinitrophenyl (DNP)-labeled α-casein peptides mixed with anti-DNP. Further, we incubated human plasma with a phospho-peptide library and conducted targeted analysis on EGFR and GFAP phospho-peptides. As a result, immunoaffinity against phospho-peptide GSHQIS[+80]LDNPDYQQDFFPK (EGFR phospho-site S1166) was detected in high-grade glioma (HGG) patient plasma but not in healthy donor plasma. For the GFAP phospho-sites selected, such immunoaffinity was not observed.

## 1. Introduction

Proteins in tumors may differ from proteins in normal tissue in quantity, amino acid sequence, post-translational modification or three-dimensional structure. These altered properties can potentially lead to the generation of autoantibodies [1]. Recent studies have shown that antibodies against specific tumor-associated antigens are detectable in blood in various types of cancer and could be valuable for monitoring cancer treatment [1,2,3,4,5] and, potentially, generate treatment options.

In eukaryotes, phosphorylation is a common post-translational modification in proteins. Many studies have shown that dysregulation of protein phosphorylation plays an important role in the development of cancer—as comprehensively reviewed by Ardito et al. [6] and Mahoney et al. [7]. Aberrantly phosphorylated peptides can be derived from these dysregulated cell signaling pathways in various cancers and may serve as tumor-specific antigens [7,8]. Antigenic peptides can bind to major histocompatibility complex (MHC) class I and II molecules. MHC-restricted phospho-peptides might be promising targets for cancer immunotherapy [7,8,9,10].

Developments in high-resolution mass analyzers have led to progress in targeted mass spectrometry (MS) methods, such as parallel reaction monitoring (PRM) [11,12]. PRM enables absolute and relative quantification of peptides, including phospho-peptides, with high selectivity and sensitivity [13,14]. Mapping of phospho-sites and quantification of the ratio of phosphorylation is possible in both biological and clinical samples, such as fresh-frozen specimen, formalin-fixed paraffin-embedded (FFPE) tissues, cell line cultures and body fluids [15,16,17].

Several techniques are available to purify immunoglobulins (IgG) from plasma or other body fluids; e.g., ammonium sulfate precipitation and affinity purification using protein A, protein G or ion exchange chromatography [18,19]. In contrast, Melon Gel resin (Thermo Fisher Scientific, Waltham, MA, USA) retains non-IgG proteins and hence allows enrichment of IgG directly from the sample without an extra (acidic) elution step. In this assay, we used this special property of Melon Gel resin to enrich Ig and Ig–peptide complexes that we formed by mixing clinical plasma samples with GBM-tissue-derived peptide libraries.

The aim of the present study was to develop a method to determine the immunoaffinity of plasma IgG against peptide antigens. We evaluated the applicability of this method by detecting the presence of IgG against a tumor-specific EGFR phospho-peptide in plasma from glioma patients.

## 2. Results

### 2.1. Detection of Anti-DNP-Bound Peptides with Melon Gel Resin

The feasibility of the Ab–peptide binding assay was first tested by binding and detection of DNP-labeled peptides in the presence of anti-DNP. Selectivity was determined on the basis of both unspecific binding of peptides in the absence of anti-DNP (negative control experiment) and the detection of unlabeled peptides. In both experiments (presence and absence of anti-DNP), Avastin was present as a DNP-unspecific antibody, at a 100-fold higher (based on vendor’s specifications) amount than the amount of anti-DNP when present. Selectivity was assessed by comparing the abundances of the individual peptides in the IgG-bound fraction to the abundances of the unbound and filter-bound fractions. In addition to the fractions collected during the assay described in Figure 1, in the feasibility assay we collected two filter-bound fractions, namely the peptides that passed the MW cut-off filter after acidification (FB1) and the peptides retained in the filter device (FB2).

IgG-bound, unbound and filter-bound fractions were analyzed by PRM measurements, whereby 19 alpha-casein peptides (7 of them DNP-labeled) were detected in at least one fraction. The four most abundant DNP-peptides (DNP modifications indicated with the delta mass [+166]) were TK[+166]VIPYVR, LTEEEK[+166]NR, HIQK[+166]EDVPSER and EK[+166]VNELSK, which accounted for 99.0% of the total number (MS response) of DNP-labeled peptides. Further, DNP-labeled peptides were predominately (83.4%) found in IgG-bound (IB) fractions when anti-DNP was present. When no anti-DNP was present, DNP-peptides were found mainly in the unbound (20.3%) and filter-bound (77.9%) fractions, and only to a minor extent in the IgG-bound fraction (1.8%). Unlabeled peptides were, independent of the presence or absence of anti-DNP, predominantly found in unbound (37.8% in the presence and 23.7% in the absence of anti-DNP) and filter-bound (61.5% and 71.4%, respectively) fractions. In IgG-bound fractions, only small numbers of unlabeled peptides were measured (in the presence of anti-DNP = 0.7% and in the absence of anti-DNP = 1.0%) (Figure 2 and Appendix A).

### 2.2. Selection of GBM-Associated Phospho-Peptide Antigens

In a previous study [15], we measured the total proteome and phospho-proteome of three GBM and three normal brain tissue (NBT) samples using shotgun proteomics with the aim of demonstrating the phosphorylation ratio in clinically relevant FFPE samples. In total, the bottom-up shotgun analysis of these six samples yielded 7020 phosphorylated, 1234 double-phosphorylated and 252 triple- or higher phosphorylated peptides. Comparing GBM and NBT, 238 of these phospho-peptides were exclusively found in all (3/3) GBM samples, and 2083 phospho-peptides were exclusively found in 2/3 of GBM samples. For the following Ab–peptide binding assay, we considered peptides exclusively found in at least two GBM samples—and thus not in any normal brain tissue—as putatively related to GBM and hence a potential target peptide antigen. Next, we linked the corresponding precursor proteins of these peptides to a well-known association in GBM and, as a result, selected seven GFAP peptides and one EGFR peptide as targets for the Ab–peptide binding assay (Appendix A) [20,21,22,23].

GFAP was the protein with the second highest abundance (spectra counts) in the dataset and was found in all GBM and normal brain tissue (NBT) samples with sequence coverages from 58.8% to 85.6%. Numbers of phospho-sites determined were on average five times higher in GBM samples compared to normal brain tissue (GBM: 13.3 vs. NBT: 2.7). Interestingly, peptides that cover the first 41 amino acids (within the head domain, amino acids 1–72) and the corresponding phospho-sites were exclusively identified in GBM samples. For the Ab–peptide binding assay, two phospho-peptides of the head domain, three phospho-peptides of the rod domain, and two phospho-peptides of the tail domain were selected as potential epitope targets. Compared to GFAP, EGFR was found in lower abundance (abundance ranked by spectra count of 1089 of 3236 total proteins identified). It was almost exclusively identified in two of the three GBM samples, with sequence coverages from 7.9% to 35.2% and with a total of 17 phospho-sites. The identified phospho-peptides and non-phospho-peptides of EGFR and GFAP are shown in Figure 3.

### 2.3. Ab–Peptide Binding Assay on EGFR and GFAP Phospho-Peptides

Performing an Ab–peptide binding assay using HGG and healthy plasma samples and the GBM phospho-peptide library, we detected EGFR phospho-peptide GSHQIS[+80]LDNPDYQQDFFPK (phosphorylation is annotated with the delta mass, [+80]) in three of four IgG-bound fractions of samples incubated with HGG plasma, but not in the corresponding unbound fractions. This result indicates the presence of autoantibodies against EGFR phospho-peptide GSHQIS[+80]LDNPDYQQDFFPK in these three GBM patients. The opposite result was obtained in the donor sample, where the peptide was detected almost exclusively in the unbound fraction (Figure 4). With the use of Avastin as negative control instead of plasma, the EGFR peptide GSHQIS[+80]LDNPDYQQDFFPK was solely found in the unbound fraction. In two HGG samples, pnop-13 and pnop-30, major proportions (77.9% and 80.1%, respectively) of the peptide were found in the filter-bound fraction. In one of the four samples, pnop-17, the peptide could not be detected in any of the fractions, and consequently a reliable interpretation of the result was not possible for this specific sample; hence, this sample was not further analyzed.

Furthermore, the IgG binding affinity of the seven GFAP phospho-peptides (Appendix A) was tested analogously to the EGFR phospho-peptide binding assay in five HGG plasma samples and two healthy donor plasma samples. Peptides that were not detected in any fraction (LGPGTRLS[+80]LAR, KIES[+80]LEEEIR, QLQS[+80]LTCDLESLR and ITIPVQT[+80]FSNLQIR) were excluded from further analysis. Three peptides, EAAS[+80]YQEALAR, SVS[+80]EGHLK and RS[+80]YVSSGEMMVGGLAPGR, could be detected reliably in at least one fraction of the assay, whereas peptides were not detected in the IgG-bound fraction in any of the samples, be it HGG or donor plasma. This indicated no detectable quantities of autoantibodies against these three GFAP phospho-peptides.

### 2.4. Mapping of Background and Unspecific Binding Peptides

To determine the background of the Ab–peptide binding assay, we analyzed the IgG-bound fractions (IB) of a total peptide library, in which 4256 peptides were identified and quantified. This library was tested against three samples: (a) Avastin as a blank, (b) HGG plasma and (c) healthy donor plasma. The three IgG-bound fractions were analyzed by an untargeted MS method (DDA), a method that allowed us to obtain a complete overview of the peptides of these samples but offers lower sensitivity than the PRM method used, as described before. Sixty-eight peptides were identified in the blank sample (Avastin) versus 92 and 97 in the HGG plasma and healthy donor plasma samples, respectively. These peptides, in total 127 unique sequences, were grouped depending on whether they were found in the IgG-bound fraction of the blank sample (they were also independently found in a plasma sample, N = 68 peptides), or solely in HGG plasma (N = 12 peptides), or solely in donor plasma (N =13 peptides) or in both plasma samples (N = 34 peptides) (Figure 5 and Appendix A). Precursor abundances of the peptides in the originating total peptide library were determined by label-free quantitation and assigned, if applicable, to the appropriate division (blank, donor, HGG or donor and HGG). Peptides found in the IgG-bound fractions (IB) were predominantly (100 of 127) also highly abundant in the peptide library (top abundance quartile), which had on average a 10–100-fold higher abundance than the mean abundance of all peptides in the library. Approximately half (56 of 127) of these peptides are derived from high- and middle-abundant plasma proteins (Appendix A).

## 3. Discussion

We have successfully described an assay to detect peptide antigens through their affinity for circulating antibodies in plasma. In this approach, Ab–peptide complexes are separated from non-IgG proteins and unbound low-molecular-weight peptides by Melon Gel IgG purification and molecular weight filtration. In a second step, these peptide antigens are dissociated from the antibodies by acidification, and the released peptides are separated from the IgG-fraction by acetone precipitation. Finally, sensitive and confident (selective) detection of peptides is accomplished by high-resolution mass spectrometry (parallel reaction monitoring). Hence, the antibodies themselves are actually not measured directly, but their presence is determined by the detection of corresponding peptide antigens. In practice, the association with a particular type of cancer or stage of disease is determined by the selected set of peptide antigens (peptide library) and antibody samples (e.g., plasma sample). We conducted two experiments to demonstrate the feasibility and applicability of the method.

The IgG purification method plays a central role in this assay, as it allows the enrichment of intact IgG–peptide complexes and ultimately the isolation of the bound peptide antigens. We used Melon Gel IgG purification as it offers several advantages over conventional immunoaffinity methods such as protein A or protein G affinity purification. The actual mechanism of Melon Gel IgG purification is not described in detail in the literature. Basically, Melon Gel binds all non-Ig proteins and allows the IgG fraction to be collected simply in the flow-through. Hence, in contrast to other techniques, an elution step that could potentially affect the integrity of the antibody is not needed, and the purified IgG fraction can be collected directly. The Melon Gel technique does not require immobilization of antibodies and avoids reduction in the site-specific accessibility dependent on the coupling conditions. The free IgG remains more accessible to an antigen than in the case of binding to an immobilized antibody. In addition, upscaling of the Melon Gel method is less limited than in the case of immobilized antibodies. Lopez and coworkers [20] have compared the protein G and the Melon Gel immunoaffinity purification methods and concluded that the elution step at low pH in the protein G affinity method can cause IgG aggregation. As a result, IgG cannot be presumed to be fully native and accessible, as is the case when Melon Gel resin is used to enrich immunoglobulins. Melon Gel IgG purification has been applied successfully to identify disease-related IgG in various clinical body fluids, such as CSF of multiple sclerosis pathology [21,22,23], serum of lung cancer patients [22] and M-protein serum levels related to a monoclonal gammopathy [23]. In addition to previous studies in which isolation of IgG was performed to identify or quantify IgG, we successfully proved the ability of Melon Gel to enrich intact antibody–peptide antigen complexes.

To examine the feasibility of the method, we first conducted an experiment in which a digest of DNP-labeled α-casein (as antigen) was mixed with anti-DNP. The formed antibody–antigen complexes (of anti-DNP and DNP-labeled peptides) were purified, after which the peptide antigens were isolated and measured. While DNP-labeled peptides were predominantly found in the IgG-bound fractions, just a relatively small proportion (<1%) of unlabeled peptides were found in the IgG-bound fraction. Similarly, when we conducted the experiment without adding anti-DNP, by far the largest number of the peptides detected (>98%) was found in the non-IgG-bound fractions. These results confirm the applicability of the method in this proof of principle. However, not all selected peptides were detected in any of the fractions. Consequentially, both IgG-bound and non-IgG-bound fractions must be analyzed to obtain reliable readouts of the assay while avoiding false-negative results. To confirm the absence of a detectable specific peptide affinity, the absence of the peptide in the IgG-bound fraction as well as its presence in the unbound-fraction should be confirmed. Correspondingly, to confirm the presence of affinity, detections in the IgG-bound fraction have to be set in relation with detected intensities in the unbound fractions. Analysis of the filter-bound fraction has shown that in addition to the peptide numbers in the IgG-bound and unbound fractions, peptides can be retained on the filter device. Probable reasons for the latter observation are adsorption at the surface of the filter device or incomplete flushing due to the design of the filter device used. To describe this more precisely, in this feasibility experiment a distinction was made between the fraction of peptides released after acidification through the filter (FB2) and peptides retained in the filter device (FB1). Peptides were found in both filter-bound fractions, regardless of whether these peptides were bound to antibodies or not. Therefore, the differentiation into the two filter-bound fractions did not provide any further information about the immuno-specificity of the peptides. In the subsequent experiments, these two filter-bound fractions were replaced with a filter-bound fraction representing a pool of FB1 and FB2. We could not determine if, in the filter-bound fraction, peptides were bound to antibodies or, though probably more unlikely, unspecifically bound to the filter. Consequentially, for the interpretation of the results of the Ab–peptide binding assay, we did not use information about the filter-bound fractions.

To determine the analytical background—in order to be able to differentiate between specific and unspecific binding of peptides—we conducted an Ab–peptide binding experiment using a total peptide library against Avastin (plasma blank), HGG plasma and healthy donor plasma. The resulting IgG-bound fractions were analyzed with the untargeted shotgun proteomics method to acquire an overview of all peptides, both unspecific and IgG-bound. Although less sensitive than the targeted PRM method, the untargeted shotgun method was most suitable for acquiring a comprehensive peptide map of the assay background. Analysis revealed that 68 peptides (out of a total of 4256 peptides in the peptide library) were found in the Avastin blank sample and 59 peptides in the plasma samples. Most of these peptides (99.2%) corresponded with high- or middle-abundant plasma proteins or had high abundance in the peptide library (highest quantile). Therefore, we conclude that the method is sufficiently selective to prevent nonspecific quantitative elution (slip-through) of library peptides and to assess background. Furthermore, the analysis of both fractions, IgG-bound and non-IgG-bound, is a key feature of the analysis in determining high selectivity.

Next, by using plasma samples and the phospho-peptide library derived from glioma tumor tissue, we investigated if the assay was suitable to detect Ab-binding peptides. Phospho-peptides that carry tumor-specific phospho-sites can be considered a potential novel class of tumor-specific antigens. The phospho-peptide antigens have been less studied than mutated antigens [6,7,8]. Zarling et al. found that phospho-peptides are presented on various types of cancer cells and recognized by CD8+ T cells, indicating that phospho-peptide antigens are potential targets for immunotherapy.[24] Mohammed and coworkers investigated the effect of conformational changes due to phosphorylation on the antigenic identify and concluded that the phospho-peptide neoantigen RQA_V (covering LSP-1 phospho-S251) might be a valuable candidate for cancer immunotherapy [10]. Engelhard and colleagues investigated the immunogenicity of phospho-peptides from breast cancer antiestrogen resistance 3 (pBCAR3_126–134_) and insulin receptor substrate 2 (pIRS2_1097–1105_) and concluded that the specific immunogenicity observed provides a rationale for immunotherapy targeting phospho-peptides [25].

To prove the applicability of our method, we tested for the presence of antibodies against GBM-associated phospho-peptides in plasma. We selected candidate phospho-peptides antigens from two GBM-associated proteins, EGFR and GFAP, on the basis of data from our previous study that compared GBM and normal brain tissue [15]. EGFR phospho-site S1166 was exclusively detected in GBM tissue samples but not in normal brain tissue. EGFR phospho-site S1166 has been detected by mass spectrometry in various phospho-proteomics studies, listed on PhosphoSitePlus [26]. Moreover, we observed changes in the phosphorylation level for this phospho-site as a result of serum starvation, a process that triggers EGFR phosphorylation [15,16]. Assiddiq and coworkers used multiple reaction monitoring (MRM) in the H838 lung cancer cell line to study the effect of gefitinib on the phospho-sites of the EGFR protein before and after EGF treatment [27]. They concluded that phosphorylation of S1166 could have a negative effect on cell growth and proliferation [27] in this cell line. Whether phosphorylation of S1166 is also involved in growth and proliferation in glioma patients is not reported. Doll et al. detected upregulation of pS1166 (1.6-fold), among other EGFR phospho-sites, in an HRAS-NHA cell line. This cell line was used as a model for primary GBM, and the authors suggested involvement in feedback downregulation of EGFR [28]. We further compared the phosphorylation data from these two studies [27,28] to our results obtained from glioma tissue (this study and [15]). For EGFR, we also found the S/T phospho-peptide (pT693: ELVEPLT[+80]PSGEAPNQALLR) in our dataset (for patient G11 and G17) and additional S/T phospho-peptides of EGFR that were not found in the cell lines. This emphasizes possible differences in the phosphorylation state between cell lines and tumor tissue. Ideally, the effect of a treatment on autoantibodies should be investigated by a separate study, but this would come with difficulties in the clinical implementation. As an alternative, we envision an experiment whereby GBM cell cultures are exposed to patient plasma to determine a possible effect on proliferation.

Seven different GFAP phospho-peptides were selected based on the results of our previous study [15]. Additionally, the literature indicates differences in GFAP between GBM tissue and normal brain tissue, positing, for instance, alternative splicing [29] and that autoantibodies for GFAP exist in pathologies such as paraneoplasm [30]. Higher expression of GFAP in glioblastoma is associated with poor prognosis [31], but little is known about the role of specific phospho-sites of GFAP in glioma. A multitude of GFAP phospho-sites are known, especially in the head domain in which GFAP assembly is regulated by phosphorylation [32]. Interestingly, the phospho-proteomics results of our previous study [15] showed a distinct phosphorylation pattern predominantly identified in the head but also in the tail domain of GFAP.

As a result of both experiments (EGFR and GFAP), we conclude that IgG binding to the EGFR phospho-peptide exists in plasma of glioma patients but not in plasma of healthy donors, and that autoantibodies against EGFR phospho-site S1166 are associated with high-grade glioma. The result of the AB peptide assay for the GFAP peptides suggests that there is no HGG-related immune response to the GFAP phospho-peptides investigated. Only three of the seven GFAP peptide antigens could be successfully analyzed; the other four had to be excluded from analysis because they could not be detected in any of the fractions. Still, these four peptides might have been retained on the Melon Gel if they had not been bound to an antibody. This observation shows to a certain extent the limitation of the assay.

In conclusion, antibody–peptide–antigen complex enrichment with Melon Gel has been applied successfully to investigate disease-related phospho-peptides. The Melon Gel immunoglobulin purification method is a promising technique to detect antibody (IgG)-bound peptides (antigens) in complex peptide libraries. This method has the potential to detect the putative presence of autoantibodies without knowledge of the antigen. We demonstrated a proof of concept for the presence of autoantibodies against the EGFR phospho-peptide S1166 in plasma of high-grade glioma patients. This antibody–peptide binding assay could potentially be further applied for diseases such as autoimmune diseases and for other types of cancer.

## 4. Materials and Methods

Unless mentioned otherwise, chemicals were purchased from Merck Millipore (Burlington, MA, USA) and solvents for LC-MS from Biosolve (Valkenswaard, The Netherlands).

### 4.1. Biological Material

One fresh-frozen GBM tissue sample served as the basis for the peptide libraries (a total peptide library and a phospho-peptide library, clinical information in Appendix A) and seven plasma samples from high-grade glioma (HGG) patients and two from healthy donors as source of IgG (Table 1). Two of the plasma samples from HGG patients were taken from the same patient before and after surgery and further therapy. The tissue and the other plasma samples were taken from other, different persons. The use of patient and donor material was approved by the Institutional Ethics Review Board of Erasmus MC, Rotterdam, The Netherlands (MEC 221.520/2002/262; date of approval 22 July 2003, and MEC-2005-057, date of approval 14 February 2005). Patients gave written consent to the use of their tissue or plasma for research purposes.

### 4.2. Preparation of Tissue Peptide Libraries

A total peptide library is a mixture of peptides obtained after digestion of one fresh-frozen GBM tissue, and a phospho-peptide library is a mixture of phospho-peptides obtained after phospho-enrichment of digested fresh-frozen GBM tissue. The GBM tissue sample was cut in five 8 µm fresh-frozen (FF) tissue slices on a microtome and placed in a 1.5 mL tube (Eppendorf). Four hundred µL of 0.1% Rapigest (Waters, Milford, MA, USA) was added to the tissue pellet (~300 µg of total protein) and sonified for 2 min at 70% amplitude at a maximum temperature of 25 °C (Branson Ultrasonics, Danbury, CT, USA) and subsequently heated at 99 °C for 5 min.

Next, the two peptide libraries were prepared from these tissue slices as described in a previous study [15]. In brief, tissue lysate was reduced and alkylated using dithiothreitol (DTT) and iodoacetamide (IAA), respectively. Then, 10 µg of trypsin (Promega, Madison, WI, USA) was added and incubated overnight at 37 °C. After acidification (pH < 2) and centrifugation, the supernatant was stored at 4 °C. Both desalting of the tryptic digests and phospho-peptide enrichment were performed using an AssayMAP Bravo automated platform (Agilent Technologies, Santa Clara, CA, USA) according to the manufacturer’s protocols [33]. Peptide concentration of the peptide library was determined with a quantitative colorimetric peptide assay (Pierce, Thermo Fisher Scientific, Rockford, IL, USA).

### 4.3. Isolation of Immunoglobulin-Binding Peptides

In the first step of the method, plasma samples were mixed with a peptide library and incubated for 30 min at 18 °C to allow possible binding of peptides and antibodies with the corresponding affinity. Next, the non-IgG protein fraction was removed using the Melon Gel IgG Purification Kit (Thermo Fisher Scientific) according to the manufacturer’s protocol. The flow-through fraction containing the IgG and IgG–peptide complexes was loaded onto a 30 kDa-molecular-weight (MW) filter device (Amicon, Merck, Burlington, MA, USA), which had been preconditioned for 30 min with 500 µL of 0.02 µg/µL bovine serum albumin (BSA) in phosphate-buffered saline (PBS). The low MW fraction (<30 kDa), containing unbound peptides, was removed by centrifugation (15 min at 20,000× *g*), desalted (removal of PBS), resuspended in 25 µL 0.1% TFA/2% ACN) and stored at 4 °C until LC-MS analysis (“Unbound peptides”, Figure 1). The remaining high MW fraction, containing IgGs and IgG–peptide complexes, was washed four times with 500 µL of PBS, and the retained volume (approximately 20–25 µL) was pipetted into a fresh tube and acidified with 1 µL of 50% TFA to dissociate the Ab–peptide complex and to release Ab-bound peptides. Next, acetone precipitation was performed to separate peptides from IgG by addition of 10 sample volumes of ice-cold (−20 °C) acetone, precipitation at −20 °C for 2 h and centrifugation (10 min at 20,000× *g*). Subsequently, the supernatant, containing the IgG-bound peptides (Figure 1), was dried using a vacuum centrifuge (Savant SC210A, Thermo Fisher Scientific), re-suspended in 25 µL of 0.1% TFA/2% ACN and stored at 4 °C until LC-MS analysis. Additionally, the MW filter device was rinsed with 500 µL of 0.5% formic acid (FA), dried (vacuum centrifugation) and stored at 4 °C until LC-MS analysis to check for “Filter bound peptides” eluted by acidification (Figure 1).

### 4.4. Targeted Parallel Reaction Monitoring (PRM) Measurements

Mass spectrometric measurements were performed on a nano-LC (Ultimate 3000RS, Thermo Fisher Scientific, Germering, Germany) coupled to an Orbitrap tribrid mass spectrometer (Orbitrap Fusion or Orbitrap Fusion Lumos; Thermo Fisher Scientific, San Jose, CA, USA). One half of the volume of each fraction (IgG-bound, unbound and filter-bound) was loaded on a trap column (C18 PepMap, 5 µm particle size, 100 Å pore size, 5 mm × 300 μm, Thermo Fisher Scientific) and desalted for 10 min with 0.1% TFA at a flow rate of 20 µL/min. Next, analytes were eluted from the trap column and separated on an analytical C18 column (PepMap C18, 75 µm ID × 250 mm, 2 μm particle and 100 Å pore size, Thermo Fisher Scientific) using a binary gradient from 4 to 38% solvent B in 30 min, whereby solvent A consists of 0.1% formic in water and solvent B consists of 80% acetonitrile and 0.08% formic acid in water. The flow rate was 300 nL/min and the column temperature 40 °C. For electrospray ionization, nano ESI emitters (New Objective, Woburn, MA, USA) were used at a spray voltage of 1.8 kV. The targeted MS/MS mode had the following settings: 1.6 *m*/*z* isolation width, 60,000 Orbitrap resolution, 200,000 automatic gain control and 118 ms maximum injection time. Further peptide specific settings are listed in Appendix A. PRM data were processed with the software package Skyline [34]. Peak picking was revised manually, and peptide detection was considered to be valid when consistent peaks of at least three fragments were acquired. Result tables of Skyline were further analyzed and plotted with the statistical software package R [35]. The mass spectrometry proteomics data have been deposited into the ProteomeXchange Consortium via the PRIDE [36] partner repository with the dataset identifier PXD032844 and DOI 10.6019/PXD032844.

### 4.5. Data-Dependent Mass Spectrometry Measurements

For the untargeted LC-MS measurement, we used the same instrumental setting and chromatography method as for the PRM measurements. A data-dependent MS acquisition method was used with an Orbitrap survey scan (range 375–1550 *m*/*z*, resolution of 120,000, AGC target 400,000), followed by consecutively isolation (isolation with 1.6 *m*/*z*), HCD fragmentation (30% normalized collision energy) and ion-trap detection of the peptide precursors detected in the survey scan until a duty cycle time of 3 s was exceeded. Dynamic exclusion was used with 10 ppm mass tolerance and 60 s exclusion duration. The acquired fragment mass spectra were searched against the human subset of the Uniprot database (version 2020-12-12; 20,395 entries) using Mascot (version 2.3.02; Matrix Science, UK) with tryptic cleavage, two missed cleavages allowed, oxidation of Methionine (+15.995 Da) as variable modification and carbamidomethylation of Cysteine (+57.021 Da) as fixed modification, precursor tolerance of 10 ppm and fragment tolerance of 0.5 Da. Search results were post-processed with the software package Scaffold (version 5.1.0, Proteome Software, Portland, OR, USA) to merge the individual search results, conduct protein grouping and calculate protein and peptide identification confidence levels (false discovery rate <1%). The precursor intensities of the identified peptides were determined with the label-free quantitation software package Progenesis QI (Version 2.0; Nonlinear Dynamics, Newcastle-upon-Tyne, UK).

### 4.6. Feasibility Experiments

Following the scheme of the isolation of immunoglobulin-binding peptides, first a feasibility experiment to proof the concept was performed with a hapten (dinitrophenyl, DNP) bound peptide of α-casein specific for a monoclonal antibody (anti-DNP). To partially label α-casein protein with DNP, bovine protein α-casein (0.4 mM, Sigma Aldrich) was incubated with 0.4 mM di-nitrobenzenesulfonic acid in 0.15 M potassium carbonate [37,38]. This partial DNP labeling of lysine moieties was meant to prevent the formation of relatively long tryptic peptides as DNP labeling blocks tryptic cleavage. The sample was acetone-precipitated to remove detergents and reagents, and the protein pellet was digested with trypsin, as described above.

Anti-DNP (Anti-DNP Antibody, clone 9H8.1, Cat# MAB2223, Merck Millipore, Burlington, MA, USA) and Avastin, to serve as a DNP-unspecific negative control (Roche, Basel, Switzerland), were purchased and stored according to the manufacturer’s recommendations. To test the feasibility of the method, digested DNP-labeled α-casein (10 µg), 1 µg of anti-DNP and 100 µg of Avastin were combined and incubated with gentle shaking for 30 min at room temperature. Additionally, the same experiment was conducted without anti-DNP as a negative control experiment to determine unspecific binding of DNP peptides. Isolation of IgG-bound peptides and targeted PRM measurements of DNP-labeled and unlabeled peptides were conducted as described above.

### 4.7. Selection of GBM-Associated Phospho-Peptide Antigens

To select relevant phospho-peptide candidates, we reprocessed a dataset of three GBM samples and three normal brain tissue (NBT) samples from a previous study [15]; the dataset is publicly accessible on ProteomeXchange PRIDE repository with the dataset identifier PXD017943. To compare GBM and NBT, we used data of replicates of both FFPE-embedded and fresh-frozen tissue of each sample. Data were processed by Mascot MS/MS database search and Scaffold post-processing as described above. The final selection of suitable antigen candidates was based on two criteria: (a) the respective phospho-peptides were identified specifically in GBM tissue and (b) the corresponding precursor proteins had a known association with GBM.

### 4.8. Detection of Peptide–Antigen-Binding IgG in Plasma

For each analysis, 10 µg of GBM phospho-peptide library was mixed with 100 µL of plasma containing putative phospho-peptide-binding autoantibodies. As a negative control experiment, instead of plasma, 100 µg of Avastin was added to determine unspecific IgG binding. Isolation of IgG-binding peptides and targeted PRM measurements of EGFR and GFAP phospho-peptides took place analogous to the feasibility experiment described above. To survey the overall detectable peptides and to determine unspecific and specific binding, we performed three additional analyses in which 10 µg of GBM peptide library was mixed with 100 µg of Avastin, 100 µL of HGG plasma or 100 µL of normal plasma. The three samples were processed according to the protocol described above. The IgG-bound fractions and GBM total peptide library were measured by a data-dependent shotgun method described above.

## Figures and Tables

**Figure 1 ijms-23-05061-f001:**
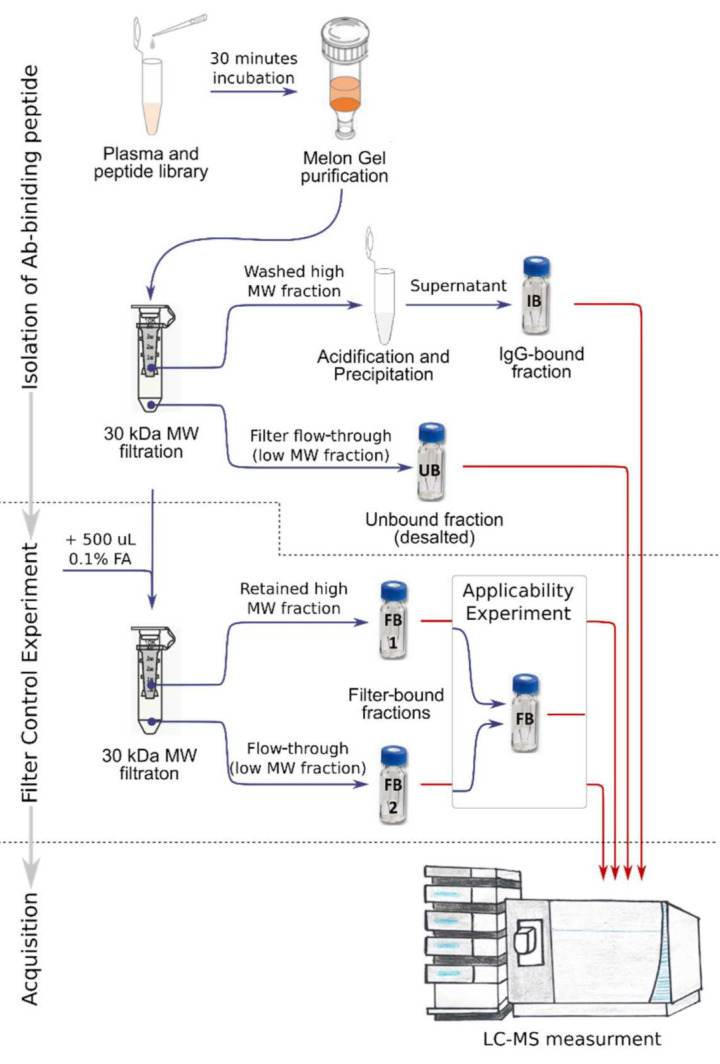
Flowchart of the antibody–peptide binding assay. IgG-bound fraction (IB), unbound fraction (UB) and filter-bound fraction (FB) annotated on LC vials are described in the text. In the feasibility experiment, two filter-bound fractions (high- and low-MW fractions) were collected and analyzed separately. In the applicability experiment, filter-bound fractions FB1 and FB2 were taken together as one filter-bound fraction (FB).

**Figure 2 ijms-23-05061-f002:**
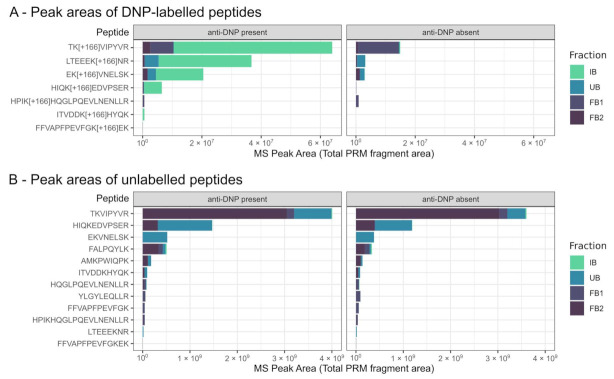
Bar chart of peak abundances of DNP-labeled peptides (**A**) and unlabeled peptides (**B**). The peptides were determined in the presence (left) or the absence (right) of anti-DNP. Color of the bars indicate corresponding fractions: IgG-bound fraction (IB), unbound fraction (UB), fraction remaining in the filter (FB1) and passed the MW filter (FB2) after acidification and centrifugation.

**Figure 3 ijms-23-05061-f003:**
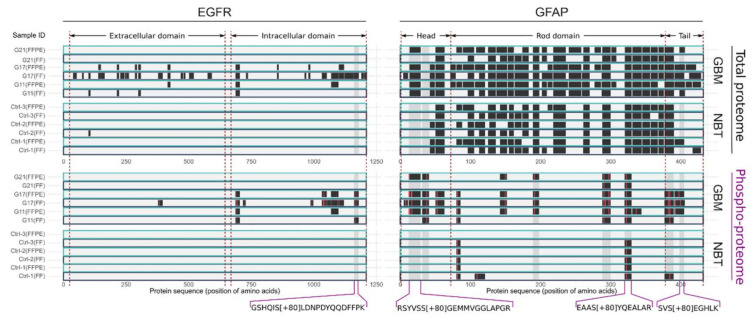
Sequence coverage and phospho-sites of EGFR (**left**) and GFAP (**right**) identified in the total proteome analysis (**top**) and phospho-proteome analysis (**bottom**) of 3 glioblastoma multiforme (GBM) and 3 normal brain tissue (NBT) samples. Results were derived from a publicly available dataset of a previous study (PXD017943). Each sample was prepared and processed in two forms, as fresh frozen tissue (FF, dark blue) and formalin-fixed paraffin-embedded tissue (FFPE, light blue). Dark segments indicate regions covered by peptide identifications, and the red lines mark identified phospho-sites. Gray regions in the sequence mark phospho-peptide targets used for PRM measurements. The four peptides that are annotated at the bottom of the plot were used as epitope-carrying peptide antigens for the Ab–peptide binding assay.

**Figure 4 ijms-23-05061-f004:**
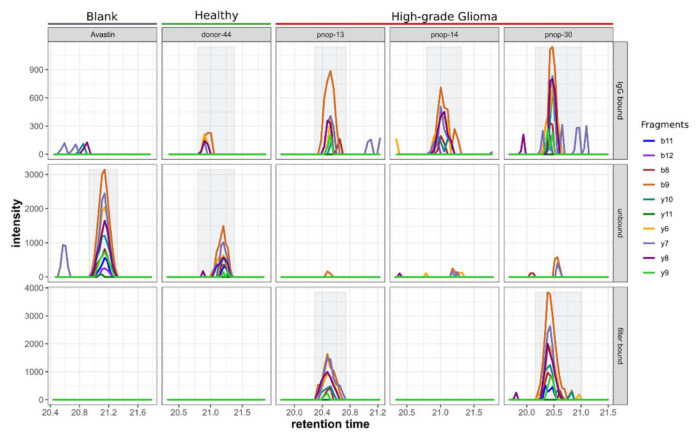
PRM peaks of EGFR peptide GSHQIS[+80]LDNPDYQQDFFPK in the IgG-bound, unbound and filter-bound fractions of one blank sample (Avastin), one healthy control plasma sample and three HGG plasma samples. A fourth HGG plasma sample did show any detectable response of this peptide and was excluded from the analysis (data not shown in figure). Colors in the chromatogram plots indicate the various fragments detected and used for quantification. Gray peak background indicates if a positive peak detection on the basis of at least 3 fragments was achieved.

**Figure 5 ijms-23-05061-f005:**
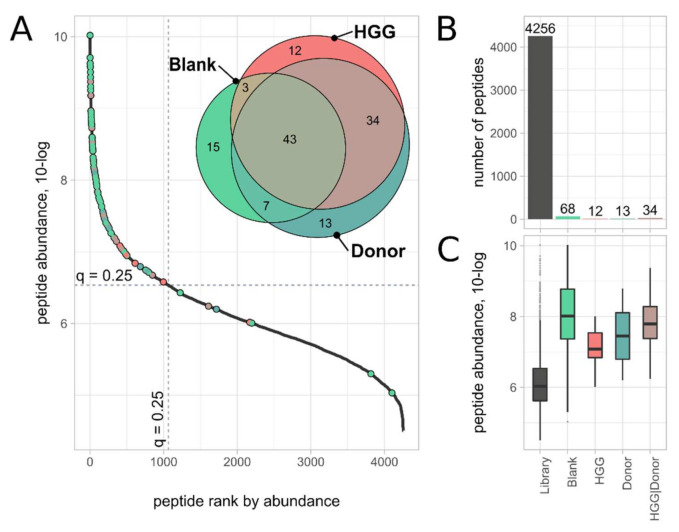
(**A**) Waterfall plot of total peptide library, with (logarithmic) peptide abundance plotted against the peptide abundance rank; peptides identified using shotgun proteomics in the blank sample (Avastin), HGG plasma or donor plasma samples are marked as points, with a color corresponding to the originating sample. Dotted horizontal and vertical lines show the abundance level (6.54) and rank (1065), respectively, of the top peptide abundance quartile. The Venn diagram insert shows the originating sample of the 127 peptides. (**B**) Bar chart of number of peptides identified in the peptide library (gray) and the Ab-binding fraction of an Avastin blank (including peptides that were found in blanks and other samples), donor plasma, HGG plasma and in both, HGG and donor plasma. (**C**) Box-plot of peptide abundances (10-log transformed) of the sets of peptides specified in panel B.

**Table 1 ijms-23-05061-t001:** List of plasma samples with clinical data and corresponding use in experiments.

Sample ID	Individual	Age	Gender	Treatment	Molecular Diagnosis	Tumor Type and WHO Classification	Experiment
Donor-44	D1	53	Male	-	-	Healthy	GFAP and EGFR
Donor-45	D2	52	Male	-	-	Healthy	GFAP
Pnop-01	P1	61	Male	Before	IDH-mutant p1p/19q No EGFR amplification	Oligodendroglioma (grade 2)	GFAP
Pnop-02	P1	61	Male	After surgery and therapy	IDH-mutant p1p/19q No EGFR amplification	Oligodendroglioma (grade 2)	GFAP
Pnop-09	P2	47	Female	Before	IDH-mutant No EGFR amplification	Glioblastoma (grade 4)	GFAP
Pnop-13	P3	75	Male	Before	Not available	Glioblastoma (grade 4)	GFAP and EGFR
Pnop-14	P3	75	Male	After surgery and therapy	Not available	Glioblastoma (grade 4)	EGFR
Pnop-17	P4	66	Male	Before	Not available	Glioblastoma (grade 4)	EGFR
Pnop-30	P5	41	Female	Before	p1p/19q	Oligodendroglioma (grade 3)	GFAP and EGFR

## Data Availability

The mass spectrometry proteomics data have been deposited into the ProteomeXchange Consortium via the PRIDE partner repository with the dataset identifier PXD032844 and DOI 10.

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
