# Peer review of "Novel Antibody–Peptide Binding Assay Indicates Presence of Immunoglobulins against EGFR Phospho-Site S1166 in High-Grade Glioma"

_ijms, 2022, doi:10.3390/ijms23095061_

Round 1
Reviewer 1 Report
The review of manuscript entitled: Novel antibody-peptide binding assay indicates presence of immunoglobulins against EGFR phospho-site S1166 in high-grade glioma. The manuscript is interesting but some things need to be improved.
First, formal requirements, literature is in the wrong format. The line numbers are removed (I suspect on purpose to make the work of the reviewer difficult). The first two pages contain empty space (also page 5) which shows sloppy manuscript preparation and disrespect for readers and reviewers.
The abstract is well prepared, however the introduction to the manuscript is not.
The introduction to the manuscript should end with the statement, e.g. “The aim of the present study was to evaluate…”. The manuscript introduction is not a summary! We do not describe the results in it, the last paragraph must be removed and corrected.
The figures are too small and the writing is heavy to read.
Did the authors include/exclude elastin derived peptide in the research? Their number increases with the age of patients, also physiologically.
Author Response
Response to the review
We would like to thank the reviewer for the comments. We have tried to address all points of the reviewer in a systematic way and have indicated where changes were made in the manuscript.
Reviewer 1
Comments and Suggestions for Authors
The review of manuscript entitled: Novel antibody-peptide binding assay indicates presence of immunoglobulins against EGFR phospho-site S1166 in high-grade glioma. The manuscript is interesting but some things need to be improved.
First, formal requirements, literature is in the wrong format. The line numbers are removed (I suspect on purpose to make the work of the reviewer difficult). The first two pages contain empty space (also page 5) which shows sloppy manuscript preparation and disrespect for readers and reviewers.
We have changed the literature format to the requirements of the journal.
We uploaded documents with line numbers. The line numbers are missing in the pdf document that was generated by the submission system. We regret that this caused difficulties for the reviewer, but it was definitely not our intention to make the work of the reviewer more difficult. In the track change Word document line numbers are available.
The abstract is well prepared, however the introduction to the manuscript is not.
The introduction to the manuscript should end with the statement, e.g. “The aim of the present study was to evaluate…”. The manuscript introduction is not a summary! We do not describe the results in it, the last paragraph must be removed and corrected.
We have edited the final part of the introduction as suggested by the reviewer (Page 4, lines 70-73).
“The aim of the present study was to develop a method to determine immunoaffinity of plasma IgG against peptide antigens. We evaluated the applicability of this method by detecting the presence of IgG against a tumor-specific EGFR phospho-peptide in plasma from glioma patients.”
The figures are too small and the writing is heavy to read.
Apparently, in the converted pdf document generated by the submission system the resolution of the figures dropped in resolution. The original uploaded figures have a high resolution (pdf vector graphics) including easy to read text labels. For your convenience we have added a Word document with track changes and including high-resolution figures.
Did the authors include/exclude elastin derived peptide in the research? Their number increases with the age of patients, also physiologically.
We didn’t include elastin-derived peptides in our research but for sure an interesting possibility for future glioma research.

Reviewer 2 Report
- This is an interesting and clinical usefully strategy to identify specific autoantibodies against potential tumor-associated peptide antigens.
- The results provided in the manuscript are convinced. I suggested defining and explaining the GBM-derived peptide library in the part of materials and methods.
- Previously, a study published in J Proteome Res. 2012 Aug 3;11(8):4110-9.demonstrated that EGFR S1166 phosphorylation represents a regulatory site that exerts a negative regulation on growth and proliferation of cancer cells. I thought the therapeutic treatment may interfere the composition or the dynamic change of autoantibodies in patient plasma, it is worthy to make a discussion of this concept.
Author Response
Response to the review
We would like to thank the reviewer for the excellent comments. We have tried to address all points of the reviewer in a systematic way and have indicated where changes were made in the manuscript.
Reviewer 2
- This is an interesting and clinical usefully strategy to identify specific autoantibodies against potential tumor-associated peptide antigens.
Thank you very much.
- The results provided in the manuscript are convinced. I suggested defining and explaining the GBM-derived peptide library in the part of materials and methods.
It has been changed as described below and changed in the manuscript, Page 17, lines 381-384
“Preparation of tissue peptide libraries
A total peptide library is a mixture of peptides obtained after digestion of one fresh-frozen GBM tissue and a phospho-peptide library is a mixture of phospho-peptides obtained after phospho-enrichment of digested fresh-frozen GBM tissue.”
- Previously, a study published in J Proteome Res. 2012 Aug 3;11(8):4110-9.demonstrated that EGFR S1166 phosphorylation represents a regulatory site that exerts a negative regulation on growth and proliferation of cancer cells. I thought the therapeutic treatment may interfere the composition or the dynamic change of autoantibodies in patient plasma, it is worthy to make a discussion of this concept.
The effect of treatment was not the primary subject of this research, but for sure of interest. Treatment of patient P3 (sample pnop-14) did not interfere with the production of autoantibodies against pS1166. It is quite difficult for us to understand how the regulation of autoantibodies in glioma takes place. Ideally, the effect of a treatment on the autoantibodies should be investigated by a separate study, but would come with difficulties in the clinical implementation. As alternative, we would envision an experiment, where GBM cell cultures are exposed to patient plasma to determine possible effects on proliferation.
Triggered by your question, we further compared the phosphorylation results of the cell lines mentioned in Assiddiq et al., 2012 and Doll et al., 2017 to our results obtained from patient material G11, G17 and G21. The publication from Doll et al., 2017, where also an upregulation of pS1166 was reported, was added to the discussion section. For EGFR, we found also the other indicated S/T phospho-peptide (pT693: ELVEPLT[+80]PSGEAPNQALLR) also in our dataset (for patient G11 and G17). Interestingly, we found also additional S/T phospho-peptides of EGFR in our dataset that were not reported in the cell line publication (Assiddiq et al., 2012 and Doll et al., 2017) emphasizing possible differences of the phosphorylation state between cell lines and tumor tissue.
We have added the discussion about this point to the body text. (Page 15, lines 328-339).
“Doll et al. detected upregulation of pS1166 (1.6-fold), among other EGFR phospho-sites, in a HRAS-NHA cell line. This cell line was used as a model for primary GBM, and the authors suggested involvement in feedback downregulation of EGFR [28]. We further compared the phosphorylation data from these two studies [27,28] to our results obtained from glioma tissue (this study and [15]). For EGFR, we also found the S/T phospho-peptide (pT693: ELVEPLT[+80]PSGEAPNQALLR) also in our dataset (for patient G11 and G17) and additional S/T phospho-peptides of EGFR that were not found in the cell lines. This emphasizing possible differences of the phosphorylation state between cell lines and tumor tissue. Ideally, the effect of a treatment on autoantibodies should be investigated by a separate study, but would come with difficulties in the clinical implementation. As alternative, we would envision an experiment, whereby GBM cell cultures are exposed to patient plasma to determine a possible effect on proliferation.”

Round 2
Reviewer 1 Report
The authors applied my suggestions